# Circulation of hydraulically ponded turbidity currents and the filling of continental slope minibasins

J. Kevin Reece ●[1] ✉, Robert M. Dorrell ●[2] & Kyle M. Straub ●[1]

Natural depressions on continental margins termed minibasins trap turbidity currents, a class of sediment-laden seafloor density driven flow. These currents are the primary downslope vectors for clastic sediment, particulate organic carbon, and microplastics. Here, we establish a method that facilitates long-distance self-suspension of dilute sediment-laden flows, enabling study of turbidity currents with appropriately scaled natural topography. We show that flow dynamics in three-dimensional minibasins are dominated by circulation cell structures. While fluid rotation is mainly along a horizontal plane, inwards spiraling flow results in strong upwelling jets that reduce the ability of mini-basins to trap particulate organic carbon, microplastics, and fine-grained clastic sediment. Circulation cells are the prime mechanism for distributing particulates in minibasins and set the geometry of deposits, which are often intricate and below the resolution of geophysical surveys. Fluid and sediment are delivered to circulation cells by turbidity currents that runup the distal wall of minibasins. The magnitude of runup increases with the discharge rate of currents entering minibasins, which influences the amount of sediment that is either trapped in minibasins or spills to downslope environs and determines the height that deposits onlap against minibasin walls.

Density driven geophysical flows help sculpt the land- and sea-scape of Earth and other planetary bodies[1–5]. Turbidity currents, a class of gravity flows that gain excess density by suspension of sediment, are the primary particulate transport process on the slope of Earth's continental margins[6]. These flows represent geohazards to submarine infrastructure[7] and transport to the deep marine huge volumes of clastic sediment[8], particulate organic carbon[9–12], and microplastics[13], in addition to dissolved nutrients and pollutants[13,14]. Models of these flows often assume that the mechanics of sediment transport by turbidity currents are similar to rivers, but recent work highlights challenges in porting knowledge from rivers to the deep marine[15]. For example, on their path down slope, turbidity currents encounter, interact with, and construct topography. Turbidity current interactions with topography can be quite complex and different than terrestrial flows, due to their relatively low contrast in density with sea water. Studies which explore the interaction of turbidity currents with submarine channels[16–19], topographic slope-breaks[20–22], and obstacles such as sea mounts[23] document how subtle topography can warp the structure of the velocity and sediment concentration fields, impacting their sediment transport capacity. Possibly the most complex interactions develop when flows enter enclosed depressions. Depressions with depths comparable to, or significantly greater than, the heights of turbidity currents enhance deposition of particulates and are termed minibasins[24].

Unfortunately, there is a lack of direct observations of field scale turbidity currents interacting with minibasins, primarily due to their: (1) relatively inaccessible locations, (2) unpredictable flow occurrences, and (3) high flow shear stresses that can destroy equipment[25,26]. Development of theory in these settings has thus leveraged numerical and physical experiments[27–32]. Due to computational demands, many

[1]Department of Earth and Environmental Sciences, Tulane University, New Orleans, LA 70118, USA. [2]Energy and Environment Institute, University of Hull, Hull HU6 7RX, UK. ✉e-mail: jreece@tulane.edu

numerical models utilize depth-averaged flow parameters[33,34]. These models afford some insights in unconfined settings and help to define the link between fluid and sediment transport dynamics and the shape of submarine fans[35]. However, their depth-averaged formulations limit their applicability in settings where vertical flow properties vary strongly in space and time, such as in minibasins[27,36,37]. In addition, while a few physical experiments document flow interactions with topography in three-dimensions, 3-D[31,38,39], most physical experiments on turbidity current – minibasin interactions have been conducted in 2-D[29,37,40–42]. Further, most of these experiments utilize quartz particles with grain sizes that are difficult to keep in suspension at laboratory scales.

Minibasins are efficient traps of particulate material because they can induce hydraulic flow ponding, a process that initiates with the reflection of flows off confining topography. This triggers a rapid spatial flow deceleration to extremely low densimetric Froude, $Fr_d$, conditions and the formation of a placid flow transition with the overlying ambient fluid, decreasing the entrainment of ambient fluid into the current[40,43]. Flow circulation within 2-D minibasins was documented along a vertical plane, with a return flow positioned above down-basin directed flow[37] (Fig. 1A). Below the return flow, ponding leads to concentration profiles with little vertical structure, as sediment lost to deposition is replaced from above with more sediment laden flow. This produces tabular deposits that do not rapidly thin against confining topography[29,40]. Previously it has been unclear if this style of circulation develops in 3-D minibasins. However, flow circulation in minibasins has implications for the near bed shear stress and sediment transport capacity, thus impacting the trapping potential of particulates.

Here, the influence of flow discharge into circular minibasins on the 3-component velocity field, structure of sediment concentration profiles, and resulting turbidite shape is studied. To accomplish this, a turbidity current mixture is developed that overcomes many past experimental scaling challenges. Further, turbidity currents are released into minibasins with scales that surpass prior experimental setups. This scale is sufficient to allow the geometry of experimental minibasins to resemble their field scale counterparts, while also trapping flows of sufficient thickness to allow for measurement of flow structure. Flow discharge is varied by adjusting input flow width, while keeping all other input conditions constant (flow height, mean inlet velocity, sediment concentration). The setup is designed to capture end members across a spectrum of complete flow trapping, stripping of the upper current, and focused discharge of the full flow to down-slope environs (i.e., the fill-to-strip-to-spill transition)[44–48]. Lateral circulation cells are discovered and quantified, which are the primary process responsible for distributing sediment throughout minibasins. Further, these circulation cells link to upwelling flow that impacts sedimentation processes by countering the still fluid settling velocity of particles. This upwelling has particular importance for the trapping of particulates with low settling velocities (i.e., particulate organic carbon and microplastics).

## Results
### Experimental design
Turbidity currents were released into minibasins with a 3 m diameter. This scale is significantly greater than prior experimental campaigns that quantified the fluid dynamics of ponded flows (e.g., Bastianon[49] where basin diameter = 1 m with ~40% sidewall slopes and Maharaj[39] where basin diameter = 0.5 m with ~30% sidewall slopes). These prior experiments had minibasin side wall slopes that were significantly steeper than observed in nature, where slopes rarely exceed 14%. The steep slopes allowed minibasins to achieve significant depth with minimal planform diameter, which aided monitoring of flows that had thicknesses comparable to minibasin relief. The circular minibasin in this study had a 10% sidewall slope and a 0.12 m depth. Dimensionless ratios characterizing minibasin topography, including side wall slope, fall within distributions generated from 2,324 depressions extracted from the Bureau of Ocean and Energy Management's bathymetric dataset of the northern Gulf of Mexico[50]. Sustained turbidity currents were delivered to the rim of minibasins for 30 minutes. Input flows had

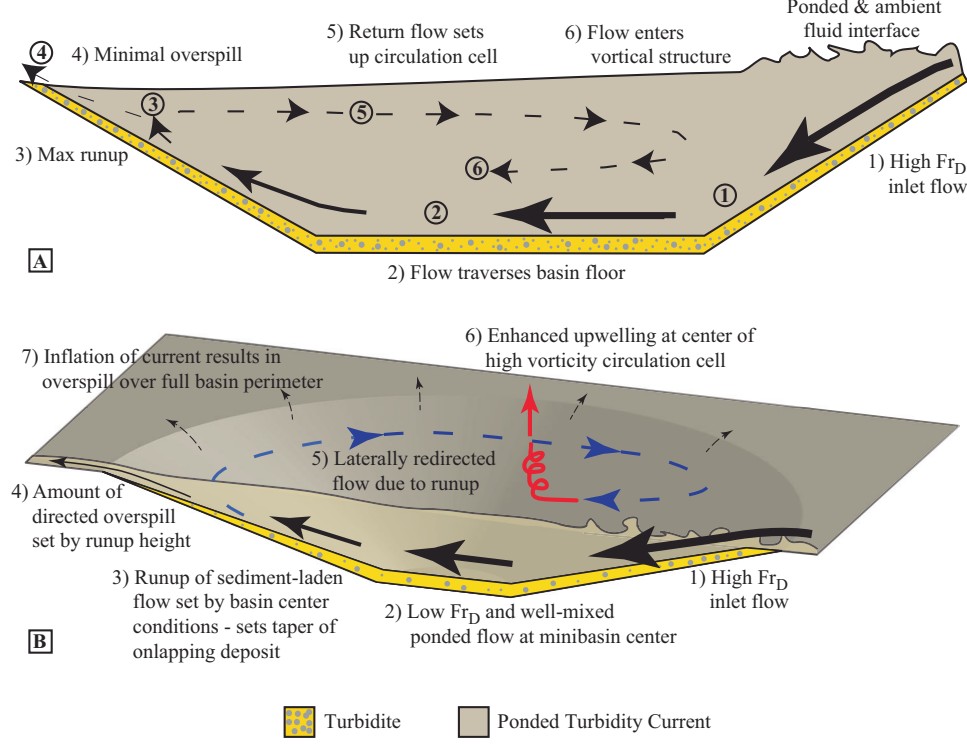

**Fig. 1 | Schematics of turbidity current – minbasin interactions. A** 2-D and (**B**) 3-D schematics of circulation cell development inside topographically enclosed minibasins.

densimetric Froude numbers of 1.1, were 48 mm thick, and had an excess density of 2.9%.

Many turbidity current physical experiments use quartz sediment[31,37,40,49], which is difficult to keep in suspension at laboratory scale, even those in the fine silt size range. To overcome this, some studies add salt to enhance the driving gravitational force and production of turbulence[17,51,52]. This is rationalized by equating the dissolved salt with washload, a fraction of the sediment load that bypasses a reach with limited to no bed interaction. Use of salt would not mimic field scale processes of turbidity current interaction with minibasins that can trap flows and all their sediment. To overcome prior experimental limitations, a slurry recipe was developed, composed of 1% aluminum oxide sediment (particle density of 3950 kg/m³ and median particle diameter of 14 μm) and a deflocculant mixture containing calcium carbonate and sodium hexametaphosphate (SHMP) that was used to inhibit amalgamation of fine scale particles. The high-density aluminum oxide sediment produces significant excess density from low volumetric sediment concentrations, generating swifter, more turbulent flows[15]. This allows transport of particles to greater distances prior to deposition.

Three experiments were performed, each composed of two flow events, and are referred to as the low-flux, mid-flux, and high-flux experiments. The range of input discharge results in flows that span the fill (low-flux) to strip (mid-flux) to spill (high-flux) spectrum. During the first event a 3-component velocity profile timeseries was collected at minibasin center for the duration of the flow using a Pulse Coherent Acoustic Doppler Profiler (PCADP), in addition to sediment concentration profiles collected after equilibrium conditions were reached. During the second event, velocity profiles were collected after equilibrium conditions were reached at a set of positions covering the river-left side of the minibasins. Topography was mapped with a displacement laser before and after each experiment.

## Minibasin center conditions

Equilibrium velocity conditions at minibasin center are estimated by averaging profiles collected from the first flow of each experiment over the duration that concentration profiles were collected. Here, $u$, $v$, and $w$ refer to the velocity components in the down-basin, cross-basin, and vertical directions, respectively. For comparison, $u$ profiles at minibasin center are normalized by the maximum down system velocity of a profile, $u_{max}$ (Fig. 2A). Profiles collected from unconfined flows typically have a velocity maximum at a height that is between 10 and 35% of the total flow height[8,52,53]. In contrast, the flow structure from the confined conditions herein display significantly elevated velocity maxima. The low-flux condition has the most complicated velocity structure, with low velocities in the lower third of the flow, peak velocities in the middle third and a rapid velocity reduction in the upper third of the flow. The mid and high-flux conditions are less stratified and have peak velocities just below the minibasin rim elevation.

Sediment concentration profiles are compared following normalization by near bed conditions, $C_{nb}$ (Fig. 2B). The low-flux experiment, which was the most contained within the minibasin, is the most stratified. The mid and high-flux conditions are well mixed in the lower two-thirds of the elevations contained within the minibasin. Sediment concentrations then rapidly decrease to near zero values approaching the rim elevation.

## Evolution of down minibasin velocity

Experiments had differences in discharge, controlled by initial flow width, that generated different minibasin floor velocity due to varying lateral flow expansion (Fig. 2C) between experiments. All experiments show a rapid spatial deceleration in $u_{max}$ with distance into the minibasin, as flow ponded triggering a rapid increase in flow height and decrease in densimetric Froude number. Minibasin floor velocities are

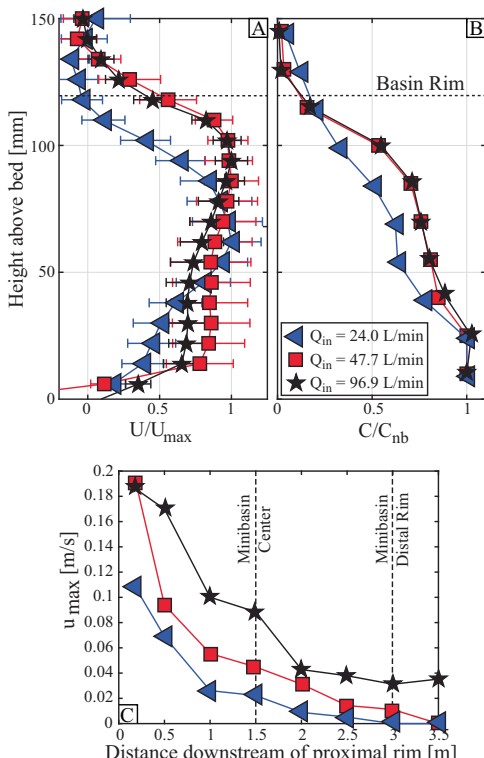

**Fig. 2 | Fluid and sediment transport fields at minibasin center.** Velocity and concentration measurements at minibasin center and flow evolution along the down basin traverse. **A**, **B** Profiles at minibasin center normalized by the maximum velocity in a profile and near bed sediment concentration, respectively. **C** Measurements of the maximum velocity along the basin bisect line.

used to estimate flow runup onto the distal minibasin wall. The magnitude of runup is estimated by:

$$\triangle z = \frac{\rho_c u_{max}^2}{(\rho_c - \rho_a)2g} \tag{1}$$

where $\rho_c$ and $\rho_a$ are current and ambient fluid densities and $g$ is gravitational acceleration[17,54]. Here, $\rho_c$ is estimated from measurements of sediment concentration. Use of Eq. 1 results in estimates of 3.9, 9.5, and 27.8 mm of runup for the low, mid, and high flux experiments, respectively. Finally, it is noted that measurements of $u_{max}$ above the downstream minibasin rim (Fig. 2C) indicate that the experiments captured the fill-to-strip-to-spill transition. The low-flux experiment (characterizing the "fill" end member) has near zero $u_{max}$ above the distal rim, which ticks up to ~15 mm/s for the mid-flux ("strip") condition and reaches ~35 mm/s for the high-flux ("spill") condition.

## Circulation Cells

Overhead imagery (Supplementary Movies 1–4) and velocity measurements covering the river left hand side of the minibasins (Figs. 3 and 4) captures paired fluid circulation cells spawned from the current interaction with the distal slope. These cells are visualized by first calculating streamlines from velocity measurements, which capture horizontal gathering of flow into the center of the circulation cells and strong upwelling flow at the cell center (Fig. 3). The circulation cells span the full extent of the ponded flow, which surrounds an inlet flow region defined by high $Fr_d$ and turbulent flow conditions[40]. The inlet flow conditions do not cover the full width of the minibasins, and therefore these circulation structures control sediment transport and delivery over the majority of the minibasin area. Fluid movement through minibasins are characterized using vector maps of the

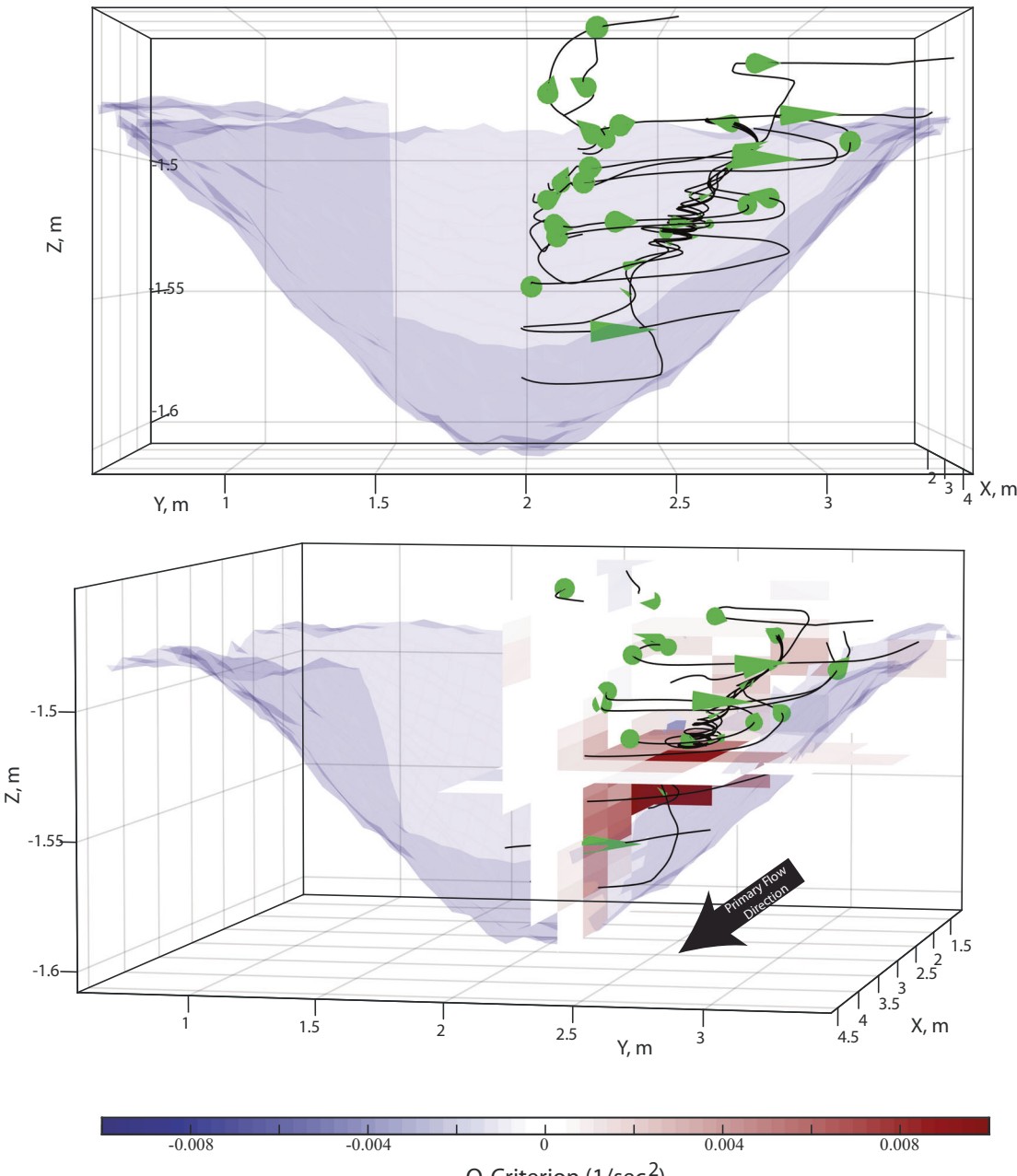

**Fig. 3 | 3-D streamlines of turbidity currents in minibasins.** Streamlines and cone plots detailing flow structure in the high-flux experiment. Minibasin topography pre-flow is illustrated in semitransparent blue mesh, with distal basin topography excluded to aid visualization. 10x Vertical exaggeration applied to aid visualization. Top panel is oriented with a view from the distal basin, looking directly upstream, while bottom panel presents a perspective view. Horizontal and vertical slices display Q-criterion. Note upwelling and spiraling current at center of circulation that corresponds to the maximum Q-criterion values.

temporally averaged depth integrated fluid flux in the down and cross basin directions:

$$q_u = \int_0^H u\,dz \qquad (2A)$$

$$q_v = \int_0^H v\,dz \qquad (2B)$$

where $H$ represents the current height, estimated with the integral length scale[55] (Fig. 4A). Temporal averaging was done over the duration that the PCADP sampled each site. When vectors are scaled by input discharge, the structure of the discharge field is remarkably similar across experiments, hinting towards the universal importance of these circulation cells for the sediment transport dynamics that control minibasin infilling across the fill-to-strip-to-spill spectrum. High fluxes down the proximal slope efficiently deliver fluid and sediment to the center of minibasins. Down basin depth integrated fluxes then rapidly decrease going up the distal minibasin slope as fluid is routed into circulation cells. Due to the inlet flow entering the center of the minibasin in these experiments, the cells are laterally offset and positioned over the lower lateral slope.

Gradients in the velocity field of the confined flow describe local fluid stretching (strain) and rotation (vorticity), which are quantified at all sample points. From Dubief and Delcayre[56], the horizontal strain

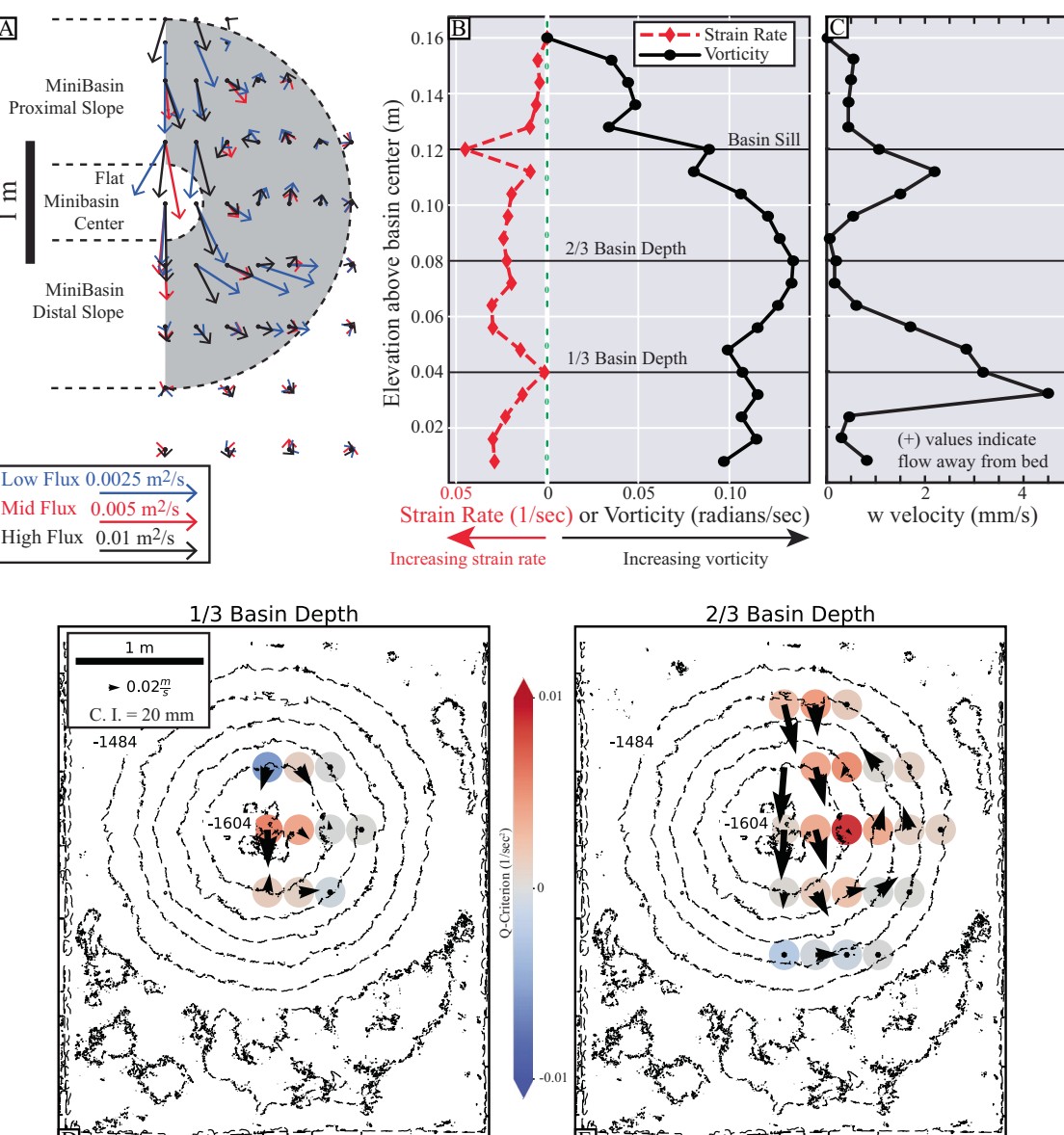

**Fig. 4 | Characterization of minibasin three-dimensional velocity field. A** Vector field of the depth integrated fluid flux with primary flow direction from top to bottom of map. **B** Magnitude of flow strain rate and vorticity and (**C**) *w* component of velocity as a function of elevation above floor of minibasin and lateral position following center of circulation cell, which migrates away from basin center with increasing flow height, as defined by the maximum Q-criterion. **C**, **D** & **E** Measurements of Q-criterion (colored dots) for the high-flux experiment along depth slices. Quivers show the *u* and *v* velocity components on each depth slice. Contours represent pre-flow topography.

rate tensor is calculated from the symmetric part of the velocity gradient tensor as:

$$S = \frac{1}{2}\left[(\delta u/\delta x) + (\delta v/\delta y)\right] \qquad (3)$$

and the horizontal vorticity is calculated from the asymmetric part of the velocity gradient tensor as:

$$\Omega = \frac{1}{2}\left[(\delta v/\delta x) - (\delta u/\delta y)\right] \qquad (4)$$

where *x* and *y* are down and cross basin locations, respectively. Strength of rotation relative to the lateral strain rate of the fluid is quantified using the Q-criterion, *Q*:

$$Q = \frac{1}{2}\left(||\Omega||^2 - ||S||^2\right) \qquad (5)$$

Positive *Q* indicates local vorticity exceeds shear (strain rate tensor), and negative values represent areas where strain rate dominates the 3-D flow field[56]. Here circulation with positive *Q* at the cell center is associated with upwelling fluid, a consequence of fluid mass conservation. This 3-D flow pattern controls sediment transport and deposition. Here, results from the high-flux condition are presented, which are similar in structure (but different in magnitude) to the other experiments (Fig. 4). Maps of *Q* at various minibasin depths highlight that the center of the circulation cells have vorticity that exceeds the strain rate (Fig. 4D, E). While *Q* values indicate whether vorticity or strain rate is larger at a point, they do not inform on the fractional difference of the two. This can be estimated with the kinematic vorticity number[56]:

$$\Omega_k^* = \frac{||\Omega||}{||S||} \qquad (6)$$

Vorticity, strain rate, and the $w$ velocity component near the center of the circulation cell are determined for all heights in the minibasin. The center of this cell laterally migrates away from minibasin center with increasing water depth (Fig. 3). Near the center of the vortex, $\Omega^*_k$ ranges between 2 to 75, suggesting limited fluid stretching during rotation (Fig. 4B). This is associated with a profile of the $w$ velocity component with upwards directed flow that considerably exceeds the still sediment fall velocity, $w_s$, of the median grain size introduced to the basin (0.5 mm/s) and the vertical detrainment velocity (Fig. 4C). This upwelling flow will influence sediment settling velocities as a function of the grain size distribution, leading to enhanced trapping potential of coarse clastic particles, relative to particulate organic carbon and microplastics that have low settling velocities. However, the profile has considerable structure with significant upwards directed flow in the lower third of the current, which reduces to near zero in the middle of the flow. This reduction might be linked to low vertical shear at the $u_{max}$ elevation[57]. The top third of the flow again is defined by upwelling that exceeds $w_s$.

## Minibasin margin onlap

Sedimentation patterns are characterized using isopach maps, calculated by differencing initial and final topography for an experiment. As a different total volume of sediment was released into the basin for each experiment, due to different flow discharges, deposition is normalized by the mean deposit thickness over the flat minibasins floor, $D^*$ (Fig. 5A–C). While the structure of the concentration profiles at minibasin center might suggest similar gradients in deposition with distance up minibasin slopes, stark differences are observed between experiments in the deposit taper against slopes. Most of the sediment released into the low-flux experiment is contained within the

minibasin, with deposit thickness at the minibasin rim only 10% of minibasin center thickness. In contrast, deposit thickness at the rim elevation exceeds 50% of minibasin center thickness for the high-flux experiment, highlighting the spilling nature of this experiment. Excluding data from the proximal slope that was covered by inlet flow conditions, the rate of thinning up minibasin slopes is quantified by binning measurements of normalized deposit thickness by elevation above minibasin center, with 1 mm tall bins. Bin averaged data generate an average onlapping profile that is a function of normalized minibasin elevation, equal to elevation above minibasin center / minibasin depth, $z^*$ (Fig. 5D). These profiles detail the rate of thinning, which is quantified with an onlap index we developed, equal to the area underneath the curves in Fig. 5D:

$$I_o = \int_0^1 D^* dz^* \tag{7}$$

Thus, sedimentation that does not change thickness up minibasin walls would yield an $I_o$ of 1, while a linear decrease in sedimentation from minibasin center values to zero deposition on the minibasin rim would yield an $I_o$ of 0.5. Here $I_o$ values of 0.48, 0.58, and 0.72 are measured for the low, mid, and high flux experiments, respectively.

## Discussion

As some of the largest sediment transport processes on the Earth's surface, turbidity currents are critically important. Traversing the seafloor they are often subject to large topographic constraints, such as minibasins. While minibasins are present on many continental margins, the northern Gulf of Mexico is characterized by an extensive

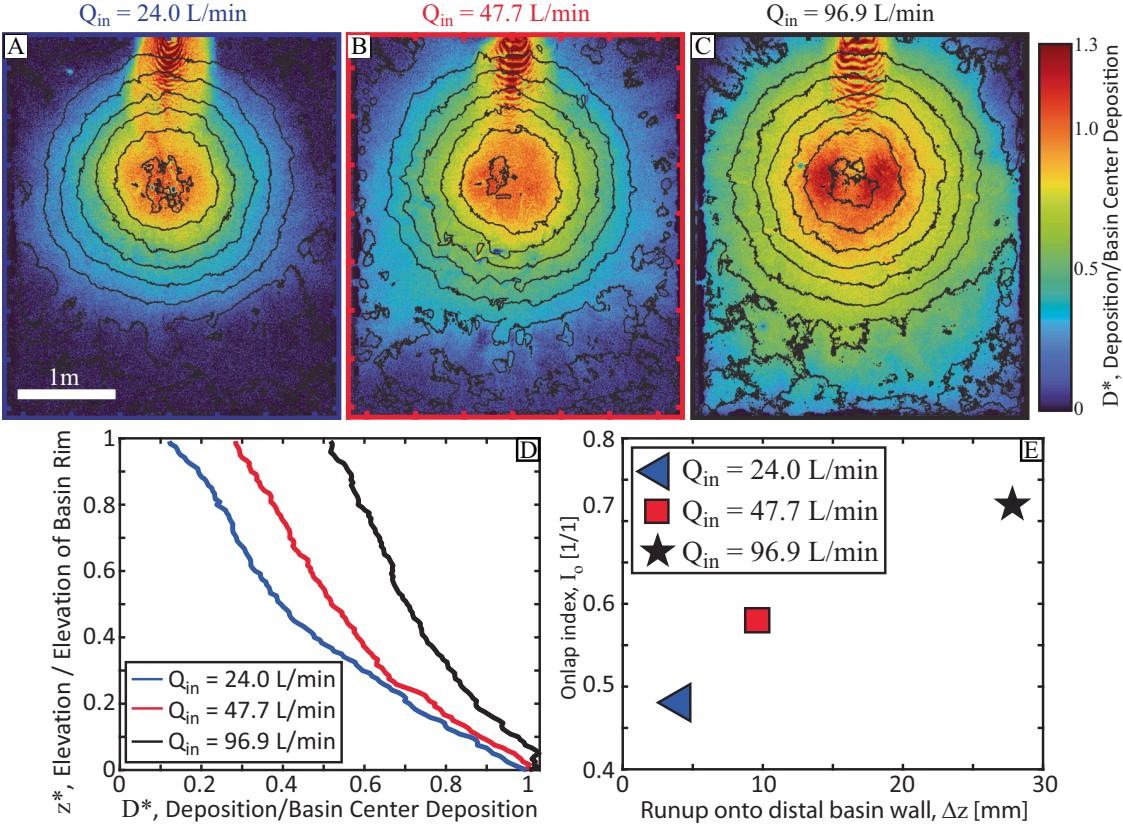

**Fig. 5 | Linking flow fields to sediment deposition. A–C** Sediment isopach maps normalized by minibasin center conditions. Contours represent initial minibasin topography. Primary flow direction in all maps is from top to bottom. **D** average sediment deposition profile up minibasin slopes. **E)** Cross-plot of estimated distal minibasin wall flow runup to onlap index.

and exquisite minibasin province resulting from the movement of the Louann salt[58,59]. The complex topography resulting from this mobile substrate and the geofluid reservoirs housed in minibasin strata have led to several conceptual models for turbidity current-minibasin interactions[46–48,60,61]. Many of these models were motivated by the Brazos-Trinity minibasin system in the northern Gulf of Mexico, which has been extensively studied through geophysical surveys and litho- and chrono-stratigraphic characterization of cores (experiments presented herein are not designed to simulate any one system and lab-field comparisons remain imperfect due to limitations in dynamic scaling methods[62]). This system also motivated earlier 2-D physical experiments, but notable differences exist in the structure of strata filling Brazos-Trinity minibasins and the deposits of these 2-D physical experiments. Namely, results from the 2-D experiments suggest that ponded turbidity currents should have limited structure to their concentration profiles, either in the vertical or down basin sense[36,63]. As a result, sustained experimental flows resulted in deposits that blanketed topography, with limited thinning over confining topography. However, deposits in Brazos-Trinity minibasins, which are thought to be the product of equilibrated and sustained flows, rapidly thin onto minibasin slopes. Further, any apparent onlap of deposits up Brazos-Trinity minibasins slopes is argued to be the product of ongoing subsidence during the last episode of sediment delivery to this system[64].

Formulations that relate slope to densimetric Froude number suggest inlet flows to many minibasins are near critical[65]. Utilizing this assumption, a comparison between the experiments described here and Brazos-Trinity Basin II is made. A portion of this minibasin's fill (Series 30) is interpreted as a ponded apron, suggesting possible hydraulic ponding conditions during deposition[60]. These turbidites have grain sizes in the mud to very fine sand spectrum. In comparison, upscaled grain sizes introduced to the experiments using established methods[66] (see extended methods) are equivalent to 73-81 μm quartz sediment in flows with a 1.8% volumetric sediment concentration, i.e. similar to those delivered to Basin II. A comparison of the ratio of input current discharge to flow trapping potential, $Q^*$, is also made for Basin II and the experiments described herein (Fig. 5). Input discharge to Basin II is calculated assuming critical $Fr_d$ conditions and estimates of flow heights and widths from the geometry of the self-formed and aggradational channel entering Basin II (see extended methods). The flow trapping potential is estimated during Series 30 as the product of minibasin area and suspended sediment settling velocity[36]. The area of Basin II at this time is estimated from published isopach maps[60], while settling velocities are calculated for the range of scaled grain sizes detailed above. This results in a possible range for $Q^*$ between 0.11 – 0.16. In the experiments $Q^*$ is equal to 0.25, 0.5, and 1.0, for the low, mid, and high flux conditions, respectively. As $Q^*$ decreased for the physical experiments, the onlap index also decreased. Carrying the near linear experimental trend between these variables to the values of $Q^*$ estimated for Brazos-Trinity Basin II, while acknowledging the limited number of experimental conditions explored, would yield an onlapping index of between only 0.44–0.46.

The link between $Q^*$ and the onlap index is hypothesized to be through the magnitude of flow runup onto distal minibasin slopes. Sediment not lost due to upwelling, flow stripping, and/or spilling ultimately gets deposited within the minibasin. Deflection of flow running up the distal slope routes sediment laden flow over the lateral minibasin slopes, resulting in deposition throughout the minibasin (Fig. 1B). It is highlighted that normalized sediment concentration profiles at minibasin center are similar for the three experimental conditions, but the onlap index, $I_o$, varies greatly between experiments (Fig. 5). However, a comparison of $\Delta z$ resulting from runup to the onlap index (Eq. 7) yields a near linear trend (Fig. 5E). This suggest that the magnitude of runup on the distal minibasin slope sets the amount of sediment delivered to circulation cells, which then distribute sediment minibasin wide. The

theory behind this prediction might therefore explain the limited amount of onlap of turbidites onto Brazos-Trinity minibasin slopes, which has been frequently noted[47,60]. This result also supports arguments that some of the apparent onlap noted in the ponded apron fill of Basin II might be the result of active subsidence during deposition[64].

A key finding of this study is the new observation of paired circulation cells resulting from turbidity currents interacting with minibasin topography (Supplementary Movies 1–4 and Fig. 1B, 3, and 4). Velocities within these circulation cells vary as a function of input discharge. However, their structure, following normalization, is remarkably similar over the fill-to-strip-to-spill spectrum (Fig. 4A). This structure is setup during the initial traverse of the turbidity current front, which does not fill the full minibasin width (Supplementary Movies 1–4). Reflection off the distal slope results in return flow along lateral minibasin slopes. This same structure is observed during equilibrium conditions, where inlet flow sends dye into the minibasin center with minimal widening until it reflects laterally when running up the distal slope (Supplementary Movies 1–4).

Prior 2-D experiments highlighted circulation in minibasins along a vertical plane[37] (Fig. 1A). During equilibrium conditions, return flow is not observed at the center of the minibasin (Fig. 2). This suggests the ability of currents to laterally expand and setup circulation along a horizontal plane suppresses the development of circulation along a vertical plane. As a result, sediment charged flow that cannot escape over the distal rim is directed to and deposited on the lateral slopes.

The study of circulation cells in flows has a long history in sedimentology, including controlling the formation of river meanders[67] and bedform development[68]. Here, the centers of minibasin circulation cells have positive Q-criterion indicating the importance of lateral fluid rotation in ponded turbidity currents. The gathering of flow towards the center of cells (Fig. 3) drives upwelling with vertical fluid velocities that exceed the still fluid settling velocity of sediment introduced to the minibasins (Figs. 3 and 4C). However, vertical velocity profiles suggest sediment entering the lower portions of the vortex might not be able to transit to the flow top. This likely creates a sediment trap that enhances sediment concentrations until the flow wanes and sediment rains to the bed. This could be the reason for the thick deposits offset either side of minibasin center in the high-flux experiment (Fig. 5C), with a counter argument being that extension of high velocity flow into the minibasin center reduced deposition rates over the minibasin floor, relative to slopes. However, sediment entering the vortex in the upper third of the flow likely can escape the flow top, reducing basin sediment trapping potential relative to theory generated from 2-D minibasin experiments[36]. Thus, circulation cells likely play a significant role in the fractionation of particulates, pollutants, and nutrients. This preferential expulsion of low settling velocity particulates likely enhances flow stripping, which has previously been linked to coarsening of proximal intraslope fans[69]. Specifically, in a linked system of minibasins, the ability for proximal minibasins to act as a sink for microplastics and particulate organic carbon, which have low settling velocities, may be significantly reduced.

The development of circulation cells and the magnitude of flow runup as a function of influx have additional potential implications for models of the temporal evolution of linked minibasins[47,48], such as the Brazos-Trinity system. Specifically, during the early filling of minibasins, development of circulation cells likely enhances the flux of sediment delivered to downslope minibasins. However, the enhanced export of fines due to the upwelling at the center of circulation cells might mean that the early fill of distal minibasins is finer than proposed by models that did not include circulation cells[47,48]. Our results also highlight that even in minibasins with significant focused flow spilling over a down-basin rim (i.e., the high-flux experiment), circulation cells develop along the lateral margins of minibasins. This circulation develops in relatively quiescent and ponded flow, in comparison to a core of higher velocity and more turbulent flow that extends from the basin inlet to distal sill.

This suggest strong lateral fining in minibasin fill during time periods of focused spilling of flow to down slope minibasins.

Finally, the vertical flow structure captured in these experiments, which is the result of flow ponding and the development of circulation cells, differs strongly from unconfined turbidity currents[52,53]. Development of rules and theory emanating from 3-D experiments will aid future development of layer-averaged models of turbidity currents interacting with complicated topography. Specifically, they offer test data to develop shape functions[70] for velocity and sediment concentration structure that could be used for improving the performance of layer average models.

## Methods
### Expanded experimental methods
Experiments were performed in a $6 \times 4 \times 2.2$ m basin. Circular minibasins were carved into 300 μm sand on a raised platform within the experimental basin that was surrounded by moats to limit current reflections off the larger experimental basin side walls. Minibasins had a diameter of 3 m, 10% sidewall slopes, and a 0.12 m depth. Minibasins were submerged in room temperature fresh water with 0.69 m of water above the minibasin rim. Turbidity currents released into the experimental basin gained excess density through suspension of aluminum oxide sediment in room temperature water with a deflocculant that consisted of calcium carbonate and sodium hexametaphosphate. The experiments used 4 grams of deflocculant for every liter of fluid. The mass in the deflocculant was 21% calcium carbonate and 79% sodium hexametaphosphate. Volumetric sediment concentration was 1% with $D_5$, $D_{25}$, $D_{50}$, $D_{75}$, $D_{95}$ of 6, 11, 14, 17, and 24 μm, respectively. Input flux was 24, 47.7, and 96.9 l/min with corresponding entrance slots that were 65, 130, and 260 mm wide for the low, mid, and high flux experiments, respectively.

The three-component velocity field was measured using A 2 MHz Nortek pulse coherent acoustic doppler profiler (PCADP). The PCADP was attached to a robotic arm on a measurement carriage suspended above the experimental basin. This carriage can move instruments to locations within the experimental basin with a 1 mm precision. The PCADP measures the velocity field once per second in a series of 8 mm tall bins beneath the probe. A profile of sediment concentrations at equilibrium conditions was collected 26–27.5 min into each experiment. The sediment concentration profile was collected with a system of ten siphons vertically stacked with 15 mm spacings and positioned at minibasin center 10 mm above the sediment interface. Following fluid evaporation, extracted sediment mass was measured to generate concentration measurements at basin center. Three cameras fixed above the experimental basin capture in high spatial resolution the evolving flow field at 0.25 Hz through the entirety of each flow event. These images were used to make time lapse videos of the experiments. A zero-offset Keyance laser, contained in a water proof casing, submerged below the water line was used for measuring topography over a 5 mm by 5 mm grid with a 0.25 mm vertical resolution. Isopachs were generated from differencing initial topographic scans before each flow event and from post-flow scans.

### Comparison and Scaling with Brazos-Trinity Minibasin II
Following established methods, we scale our experimental conditions to field conditions. We focus this endeavor on flows of the scale that filled Brazos-Trinity Minibasin II. We emphasize that our experiments were not designed to simulate any one field site and that established engineering scaling methods carry limitations[62]. As such, the scaling presented here is only intended to guide how experimental results might be applied to the interpretation of field scale minibasins. We apply a dynamic scaling protocol that assumes similarity between the model (experiment) and prototype (field) systems and focuses on the densimetric Froude number, equal to:

$$Fr_d = \frac{\bar{u}}{\sqrt{RgCH}} \tag{8}$$

Where $\bar{u}$ is the mean current velocity, $R$ is the submerged specific gravity of sediment, $g$ is gravitational acceleration, $C$ is the volumetric sediment concentration, and $H$ is the current height. We set $Fr_{d(model)} = Fr_{d(prototype)}$, which with the rules defined in Graf[66], under the constraint of constant reduced gravity (i.e., $RgC$), and a geometric scale factor, $\lambda$, results in the following relationships:

$$u_p = \lambda^{1/2} u_m, \, w_{s,p} = \lambda^{1/2} w_{s,m} \tag{9a, b}$$

We measured a 30 m deep self-formed and aggradational feeder channel to Basin II using the BOEM bathymetry[50] map, suggesting input flow heights between 30-45 m[71]. Given this range and the height of our experimental input flows, we apply a range geometric scale factors between 600 and 900, which yield estimates of input $u_p$ between 3.2 and 3.9 m/s and $w_{s,p}$ between $3.85 \times 10^{-3}$ and $4.71 \times 10^{-3}$ m/s. This range in settling velocities can be converted to a quartz particle diameter using the Ferguson and Church method[72] and suggests a prototype quartz sediment in transit between 73-81 μm, similar to the ponded apron fill of Basin II[60].

We also estimate a ratio of an input flow discharge to minibasin flow trapping potential for both our experiments and Brazos-Trinity Basin II. We estimate the discharge delivered to Basin II as the product of our estimated flow velocity, flow depth, and flow width. Flow width is estimated from a measured feeder channel width of 225 m and again we explore a range of flow heights and associated flow velocities, as outlined above, which yields a range of input minibasin discharges between $2.2 \times 10^4$-$4.0 \times 10^4$ m³/s. Next, we estimate the minibasin flow trapping potential as the product of the still fluid sediment settling velocity and the minibasin planform area[36]. We use our prototype settling velocities and an area of Basin II equal to $5.2 \times 10^7$ m², estimated from the area of Series 30 deposition within Basin II[60]. This yields a plausible range of flow trapping potential between $2 \times 10^5$-$2.5 \times 10^5$ m³/s. This then yields a range in the ratio of input flux to minibasin flow trapping potential between 0.11 and 0.16 for Basin II. This same ratio was between 0.25 and 1.0 for our suite of experiments. While not an identical match, the experimental conditions are within an order of magnitude of estimated field conditions, supporting our assertion that circulation cells are also important for distributing sediment in field scale minibasins.

## Data availability
Data that support this study can be downloaded at https://doi.org/10.5281/zenodo.8144554.

## Code availability
Study does not utilize custom code or mathematical algorithms central to the conclusions.

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

## Acknowledgements

Support from grants to Reece from The Geological Society of America, American Association of Petroleum Geology, and by SEPM, Society for Sedimentary Geology; grants to Straub from The Tulane Carol Lavin Bernick Fund, ExxonMobil, The U.S. - Israel Energy Center funded through the Binational Industry Research and Development Foundation; and grants to Dorrell by the UK Natural Environment Research Council NE/S014535/1.

## Author contributions

J.K.R. and K.M.S. conceived the initial idea of the study. R.M.D. developed the experimental sediment mixture. J.K.R. and K.M.S. lead the development of experimental matrix with input from R.M.D. Experimental protocol designed by K.M.S. with input from J.K.R. Experiments were run by J.K.R. with help from K.M.S. All authors contributed to the data analysis and interpretations. J.K.R. wrote the initial draft of the manuscript with edits provided by K.M.S. and R.M.D.

## Competing interests

We declare no competing interests.
