## [Peer Review File · Nature Communications]

Circulation of hydraulically ponded turbidity currents and the filling of continental slope minibasinsREVIEWERS' COMMENTS

Reviewer #1 (Remarks to the Author):

The authors report on a series of novel laboratory experiments that investigate the dynamics of turbidity currents that propagate along the seafloor and interact with three-dimensional minibasins. They observe the formation of circular flow cells that rotate around a vertical axis and eventually generate an upward jet of sediment-laden flow at their center.

This observation substantially modifies our previous understanding of turbidity currents interacting with minibasins, which is based on two-dimensional considerations and emphasizes the existence of a horizontal axis of rotation.

To my knowledge, this is the first observation of such circular cells, and I believe it has a potentially major impact on the field. The methodology is sound, the experiments were conducted competently, and sufficient detail is provided for readers to reproduce the work.

Hence, I recommend acceptance of the manuscript.

Reviewer #2 (Remarks to the Author):

I have reviewed the paper by Reece et al concerning the dynamics of experimental turbidity currents in minibasins – it is well-written, well-referenced, and of broad impact – I recommend minor revisions. In no particular order, my recommendations are:

1. Authors could cite a few more papers about turbidite deposition in minibasins and other complex-slope environments. Of course, I can recommend to cite my own papers on the topic (which you shouldn't feel compelled to do so), as well as papers by others. I make these recommendations in the text as comments, and include a few others here:

a. I recommend to cite a few numerical and experimental models of lobe/fan deposits, usually with Alessandro Cantelli as the first author or co-author, this work is fantastic and quite relevant to your study. The thesis of Sherif Abd-el-Gawad is a gem, FYI.

b. Jobe et al 2017 describes lobe deposits in an intraslope setting that differ from unconfined lobe deposits, and presents a model for how they are different, which matches pretty closely your observations about coarse vs fine sediment deposition <https://doi.org/10.1130/B31440.1>

c. Not sure where exactly to cite this, but Jobe et al 2018 also worked on compiling data from modern systems to look at turbidity current event sediment flux – no need to cite this, but might be an interesting read for you to consider how your scaled data may fit these observations <https://doi.org/10.1002/dep2.42>

2. Perhaps it is the formatting of the journal, but I didn't find a punchy conclusions section that wraps

everything up – is there a plan to add that? If not, you could modify the “Importance of circulation cells to trapping of particulates” section to add something like this. I feel like there could be additional focus on:

a. The coarse vs fine sediment distribution in linked minibasins – this is quite a hot topic for oil explorers, and is also quite important for predicting where fine grained stuff like carbon and plastics will be deposited. A proxy for this are your low and high flux cases, where the strip and spill are different. The info is all there, it is just making sure the reader understands the importance of your work for this type of prediction! You could also add notes in Fig 1 about what type of sediment spills and the implications for grain size/type distribution in different basins

b. I looked for a publication that focuses on this (and failed to find a good one), but there are a lot of mapped sand bodies in minibasins that are not basin-centered, but rather thicker along the edges. Everyone always attributes this to salt growth, but your data (high-flux isopach and the general circulation model) suggests that another mechanism could account for that – might be worth asking around to find a good paper to cite for this, and then including a few sentences on the topic.

c. See #4 below, could consider those ideas in this section too

3. The references are not formatted consistently, this is an easy fix during copy-editing (e.g., the Prather and Pirmez papers on the Brazos-Trinity system).

4. For the movies, it would be great to add “low flux” “medium-flux” or “high-flux” to the movies, instead of the “21-4-flow1”, which is kind of confusing. I would also love to see a single movie that has all three side by side with the timing of die injection approximately the same – it would really show the circulation cell nicely, and the differences in that cell in the three flows. In the high flux case, the die hangs around in the eddy for a while, while the main part of the basin is flushed clean, super interesting for thinking about sediment distribution related to entry points in a minibasin...

It was a pleasure to read this and I look forward to seeing it in “print” soon.

Zane Jobe

REVIEWERS' COMMENTS

Reviewer #1 (Remarks to the Author):

The authors report on a series of novel laboratory experiments that investigate the dynamics of turbidity currents that propagate along the seafloor and interact with three-dimensional minibasins. They observe the formation of circular flow cells that rotate around a vertical axis and eventually generate an upward jet of sediment-laden flow at their center.

This observation substantially modifies our previous understanding of turbidity currents interacting with minibasins, which is based on two-dimensional considerations and emphasizes the existence of a horizontal axis of rotation.

To my knowledge, this is the first observation of such circular cells, and I believe it has a potentially major impact on the field. The methodology is sound, the experiments were conducted competently, and sufficient detail is provided for readers to reproduce the work.

Hence, I recommend acceptance of the manuscript.

We very much thank the reviewer for their time assessing this submission and very much appreciate their positive evaluation.

Reviewer #2 (Remarks to the Author):

Dear Editor,

I have reviewed the paper by Reece et al concerning the dynamics of experimental turbidity currents in minibasins – it is well-written, well-referenced, and of broad impact – I recommend minor revisions. In no particular order, my recommendations are:

Thank you for this positive news

1. Authors could cite a few more papers about turbidite deposition in minibasins and other complex-slope environments. Of course, I can recommend to cite my own papers on the topic (which you shouldn't feel compelled to do so), as well as papers by others. I make these recommendations in the text as comments, and include a few others here:

a. I recommend to cite a few numerical and experimental models of lobe/fan deposits, usually with Alessandro Cantelli as the first author or co-author, this work is fantastic and quite relevant to your study. The thesis of Sherif Abd-el-Gawad is a gem, FYI.

We have added the following references that include Cantelli. These are utilized in the introduction section of the manuscript to denote how interaction of turbidity currents with topography alters flow properties.

Fernandez, R. L., Cantelli, A., Pirmez, C., Sequeiros, O. & Parker, G. Growth patterns of subaqueous depositional channel lobe systems developed over a basement with a down-dip break in slope: laboratory experiments. *Journal of Sedimentary Research* 84, 168-182 (2014).

Abd El-Gawad, S. et al. Three-dimensional numerical simulation of turbidity currents in a submarine channel on the seafloor of the Niger Delta slope. *Journal of Geophysical Research: Oceans* 117 (2012).

b. Jobe et al 2017 describes lobe deposits in an intraslope setting that differ from unconfined lobe deposits, and presents a model for how they are different, which matches pretty closely your observations about coarse vs fine sediment deposition <https://doi.org/10.1130/B31440.1>

The following sentence and reference to this paper has been added to the section on Importance of circulation cells to trapping of particulates: “ This preferential expulsion of low settling velocity particulates likely enhances flow stripping, which has previously been linked to coarsening of proximal intraslope fans⁷⁰.“

c. Not sure where exactly to cite this, but Jobe et al 2018 also worked on compiling data from modern systems to look at turbidity current event sediment flux – no need to cite this, but might be an interesting read for you to consider how your scaled data may fit these observations <https://doi.org/10.1002/dep2.42>

Thank you for passing this work on to us. We will certainly give it a read.

2. Perhaps it is the formatting of the journal, but I didn't find a punchy conclusions section that wraps everything up – is there a plan to add that? If not, you could modify the “Importance of circulation cells to trapping of particulates” section to add something like this. I feel like there could be additional focus on:

Correct, our understanding is that Nature Communication manuscripts generally do not include formal Summary or Conclusions sections.

a. The coarse vs fine sediment distribution in linked minibasins – this is quite a hot topic for oil explorers, and is also quite important for predicting where fine grained stuff like carbon and plastics will be deposited. A proxy for this are your low and high flux cases, where the strip and spill are different. The info is all there, it is just making sure the reader understands the importance of your work for this type of prediction! You could also add notes in Fig 1 about what type of sediment spills and the implications for grain size/type distribution in different basins.

Thank you for this suggestion. We have now added a paragraph on implications of our work towards linked minibasin systems.

b. I looked for a publication that focuses on this (and failed to find a good one), but there are a lot of mapped sand bodies in minibasins that are not basin-centered, but rather thicker along the edges. Everyone always attributes this to salt growth, but your data (high-flux isopach and the general circulation model) suggests that another mechanism could account for that – might be worth asking around to find a good paper to cite for this, and then including a few sentences on the topic.

Thank you for this comment. Unfortunately, we have not identified the field observations you recall.

c. See #4 below, could consider those ideas in this section too

See how this is addressed above.

3. The references are not formatted consistently, this is an easy fix during copy-editing (e.g., the Prather and Pirmez papers on the Brazos-Trinity system).

Thank you for this suggestion. We edited our references.

4. For the movies, it would be great to add “low flux” “medium-flux” or “high-flux” to the movies, instead of the “21-4-flow1”, which is kind of confusing. I would also love to see a single movie that has all three side by side with the timing of die injection approximately the same – it would really show the circulation cell nicely, and the differences in that cell in the three flows. In the high flux case, the die hangs around in the eddy for a while, while the main part of the basin is flushed clean, super interesting for thinking about sediment distribution related to entry points in a minibasin...

Changes made as suggested.

Those altered in Minibasins in line 45 of the original submission.

Depth-averaged is now appropriately used through manuscript.

Elevation range of velocity max changed to 10-35% of flow height and reference to Talling work added to original submission lines

Lines 254-258 in the original submission have been removed at the suggestion of the reviewer.

Regarding lines 330-331 of the original manuscript, we have added text to highlight the competing hypothesis the review suggests.

The reviewer suggested editing language in lines 334-336 of the original manuscript to denote that sediment leaked from a proximal basin might end up in a distal basin, for a linked system of minibasins. Changes made per suggestion.

With regards to the caption for figure 4, we believe the dominant flow direction is already easily interpreted from figure, so choose to not add an arrow as suggested.

It was a pleasure to read this and I look forward to seeing it in “print” soon.

Zane Jobe